

# Depth scaling of soil moisture content from surface to profile: multi-station testing of observation operators

Xiaodong Gao[1,2,3], Xining Zhao[1,2,3], Luca Brocca[4], Gaopeng Huo[3], Ting Lv[3], Pute Wu[1,2,3]

[1]Institute of Soil and Water Conservation, Northwest A&F University, Yangling, Shaanxi Province, China
[2]Institute of Soil and Water Consevation, CAS and MWR, Yangling, Shaanxi Province, China
[3]National Engineering Research Center for Water Saving Irrigation at Yangling, Yangling, Shaanxi, China.
[4]Research Institute for Geo-Hydrological Protection, National Research Council, Perugia, Italy

*Correspondence to*: Xining Zhao (zxn@nwafu.edu.cn)

**Abstract.** The accurate assessment of profile soil moisture for spatial domains is usually difficult due to the associated costs,
strong spatial-temporal variability, and nonlinear relationship between surface and profile moisture. Here we attempted to use observation operators built by Cumulative Distribution Frequency (CDF) matching method to directly predict profile soil moisture from surface measurements based on multi-station *in situ* observations from the Soil and Climate Analysis Network (SCAN). We first analyzed the effects of temporal resolution (hourly, daily and weekly) and data length (half year in non-growing season, half year in growing season, one year, two years and four years) on the performance of observation
operators. The results showed that temporal resolution had a negligible influence on the performance of observation operators. However, data length significantly changed the prediction accuracy of observation operators, and a two-year interval was identified as the optimal data length in building observation operators. A dataset with a two-year duration was therefore used to test the robustness of observation operators in three primary climates (humid continental, humid subtropical and semiarid) of the continental USA, with the popular exponential filter employed as a reference approach. The results
indicated that observation operators reliably predicted profile soil moisture for the majority of stations in both calibration and validation periods and performed almost equally well with the exponential filter method. This suggests that observation operators are a feasible statistical tool for depth scaling of soil moisture. The findings here have the potential to be applied in profile soil moisture prediction from surface measurements at a range of environments if the target site has long enough (two years) soil moisture observations even with coarse temporal resolutions.

# 1 Introduction

Soil moisture in the root-zone profile is a key variable that influences the agricultural, hydrological, ecological and meteorological systems of the Critical Zone. It determines water availability to crops and participates in the partitioning of available energy into sensible and latent heat and of precipitation into infiltration and runoff (Montaldo and Albertson, 2003). Continuous and accurate measurement of profile soil moisture, however, is difficult because of expensive field



measurements, strong spatial-temporal variability, and the nonlinear relationship between surface and profile soil moisture (Han et al., 2012a; Dumedah et al., 2015). In contrast, the collection of surface soil moisture data is much easier. At fine scales (e.g. field, hillslope, or small watershed), surface soil moisture can be attained from portable soil moisture sensors or derived using cosmic-ray neutron probes and ground-penetrating radar (Penna et al., 2013; Ferrara et al., 2013; Baatz et al.,

2014). At larger scales (e.g. regional and global scales), remote sensing techniques can provide surface soil moisture data at fine spatial or temporal resolution (Brocca et al., 2011; Panciera et al., 2014). However, these instruments generally measure soil moisture at relatively shallow layers. For example, microwave remote sensing methods (either active or passive) retrieve near-surface soil moisture that only reaches several centimetres beneath the soil surface (Draper et al., 2011). Therefore, it is necessary to link surface soil moisture to profile soil moisture *via* robust depth-scaling approaches.

A variety of approaches for predicting profile soil moisture from surface measurements have been proposed, ranging from simple statistical relationships to physically-based retrieval (Wagner et al., 1999). The primary methods used today can generally be classified into three different types: (1) data assimilation methods; (2) analytical methods; and (3) statistical methods. Data assimilation methods refer to techniques which incorporate surface soil moisture measurements (e.g. remote

sensing products) into physically-based hydrologic models to obtain an analysis that best represents profile soil moisture and a number of data assimilation algorithms have been developed (Evensen, 1994; Walker et al., 2002; Heathman et al., 2003; Reichle et al., 2007; Draper et al., 2011; Dumedah et al., 2015). However, its application may be constrained by the required model parameters (soil properties, vegetation features and atmospheric forcing), which are difficult to obtain at larger scales, as well as by uncertainties related to the physical description of soil hydrological processes (Albergel et al., 2008; Hu and Si,

2014). The analytical methods require fewer input parameters and are computationally more efficient than data assimilation methods. They are generally mathematically derived from physically-based relationships of water flows that include some simplification assumptions (Arya et al., 1983; Camillo and Schmugge, 1983; Wagner et al., 1999; Manfreda et al., 2014). Currently, the exponential filter method introduced by Wagner et al. (1999) is likely the most popular analytical method since it only requires one parameter, the characteristic time length (T). This method has successfully predicted subsurface

soil moisture from surface observations for multiple regions that vary in climatic and/or soil conditions (Ceballos, et al., 2005; Albergel et al., 2008; Brocca et al., 2011; Ford et al., 2014; Peterson et al., 2016). In addition to the two above methods, statistical models are also introduced to do depth scaling of soil moisture due to its simplicity and are completely data driven. These methods include linear and nonlinear regression models (Jackson, 1986; Shi et al., 2014), artificial neural network (Bono and Alvarez, 2012), and time stability analysis (Hu and Si, 2014; Gao et al., 2015) among others. However,

the existing statistical methods usually defined surface soil deeper than 20 cm even down to 40 cm which is far beyond the scope of satellite sensors. This restricts the application of statistical methods to profile soil moisture estimation because in many cases only surface measurements (≤ 5 cm) are available. Despite the existing deficiency, robust statistical methods are



still appealing in predicting profile soil moisture because of their simplicity and applicability to a wide range of environments.

The Cumulative Distribution Frequency (CDF) matching method has proven to be both efficient and widely applicable

through the construction of observation operators to adjust for the systematic differences in soil moisture that arise from different sources such as *in situ* measurements, modelled outputs and remote sensing retrievals (Reichle and Koster, 2004; Drush et al., 2005; Brocca et al., 2011; Parrens et al., 2014). The method has also been extended to the spatial upscaling of point soil moisture measurements (Han et al., 2012b) and spatial transferring of soil moisture between different areas (Gao et al., 2013). The CDF matching method is expected to be applicable to the presented study because soil moisture at various

layers can be regarded as belonging to different spatial domains or sources. In this way, profile soil moisture could be predicted by adjusting for surface soil moisture through the building of the observation operators for surface and profile data.

The presented research here is not to mask the promising of physically-based and analytical models, but aims to introduce a feasible statistical approach to give robust predictions of profile (100 cm) soil moisture from surface (5 cm) observations.

Here we tested the feasibility and robustness of observation operators built by CDF matching method in scaling surface soil moisture to profile soil moisture by using multi-station in situ soil moisture observations from the Soil Climate Analysis Network (SCAN) in the United States. The paper is arranged as follows. First, we analysed how the resolution and length of soil moisture time series affected the performance of observation operators. The results were used to choose an appropriate temporal resolution and length for the construction of observation operators. We then tested the feasibility and robustness of

observation operators under different climate regions in the continental United States.

## 2 Materials and methods

### 2.1 Soil moisture datasets

Here the feasibility and robustness of a CDF matching method in the depth scaling of soil moisture from surface to profile was tested using in situ soil moisture measurements from the Soil Climate Analysis Network (SCAN). SCAN focuses

primarily on agricultural areas in the US and consists of over 200 soil moisture monitoring stations across different climate regions, mainly serving to monitor drought and climate change. Soil moisture content at depths of 5, 10, 20, 50, and 100 cm was measured at each station with HydraProbe soil moisture sensors (Stevens Water Monitoring Systems Inc., Portland, OR). All SCAN soil moisture data, at both daily and hourly resolution, are available at the National Water and Climate Center website (www.wcc.nrcs.usda.gov/scan). A total of 12 stations were chosen for analyses according to the objectives of this

study. General information about these 12 stations is presented in Table 1. Although these data have been corrected by data managers, we were able to detect outliers in the datasets. To identify outliers in one layer, soil moisture at this depth was




linked to values at adjacent depth(s) and rainfall events. On the one hand, if soil moisture in one layer clearly increased during some period but no rainfall events occurred before and meanwhile the soil moisture in adjacent layers did not show clear increase, the soil moisture values in this layer during this period were identified as outliers. On the other hand, if soil moisture in one layer clearly decreased whereas soil moisture in adjacent layers showed no clear decrease, then these soil

moisture values were also identified as outliers. The outliers were then excluded from the analyses.

In this study, surface soil moisture refers to soil moisture in the 5 cm, and the profile soil moisture refers to that in the 0-100 cm. The profile soil moisture is a depth-weighted mean of the values in the 5 (layer 1), 10 (layer 2), 20 (layer 3), 50 (layer 4) and 100 (layer 5) cm. It is calculated as follows:

$$\theta_p = \frac{2\theta_1 L_1 + (\theta_1 + \theta_2)L_2 + (\theta_2 + \theta_3)L_3 + (\theta_3 + \theta_4)L_4 + (\theta_4 + \theta_5)L_5}{2(L_1 + L_2 + L_3 + L_4 + L_5)} \tag{1}$$

where $\theta_p$ refers to the profile soil moisture content (m$^3$ m$^{-3}$); $\theta_i$ ($i$=1, 2, …, 5) refer to soil moisture content at the $i$th layer (m$^3$ m$^{-3}$); and $L_i$ ($i$=1, 2, …, 5) refer to the depth of the $i$th soil layer (m).

**2.2 Observation operators**

**2.2.1 The CDF matching method**

The Cumulative Distribution Frequency (CDF) matching method was used to create observation operators. The observation operators were then used to predict profile soil moisture content from surface measurements by adjusting for the systematic differences between these two layers. It is important to note that the CDFs used in this study were built from soil moisture time series and assume that all soil moisture values are equally probable (Pachepsky and Hill, 2016).

As shown in Figure 1, the CDF matching method rescales the CDF of one dataset (surface soil moisture time series in this study) to match that of another dataset (profile soil moisture time series in this study). In previous studies, a third-order polynomial was usually employed in defining observation operators (Drusch et al., 2005; Han et al., 2012). Since it is the first application of this method in soil moisture depth upscaling, a pre-analysis was done to identify the optimal order by using datasets in stations of Perdido Riv Farms and Willow Wells. It showed that the RMSE values decreased clearly with

polynomial order from one to five but weak gain was obtained at higher order (Figure 2). Therefore, a fifth-order polynomial was used here considering the accuracy of fitting and the principle of parsimony. The technical procedure of this method progressed as follows:

(1) The *in situ* measured surface ($\theta_s$) and profile ($\theta_p$) soil moisture datasets were ranked.

(2) Next the differences ($\Delta$) in soil moisture between corresponding elements in the surface and profile datasets were

calculated as:





$$\Delta = \theta_s - \theta_p \qquad (2)$$

(3) A fifth-order polynomial fit was then used to quantify the relationship between $\theta_s$ and $\Delta$ as:

$$\hat{\Delta} = k_0 + k_1 \cdot \theta_s + k_2 \cdot \theta_s^2 + k_3 \cdot \theta_s^3 + k_4 \cdot \theta_s^4 + k_5 \cdot \theta_s^5 \qquad (3)$$

where $\hat{\Delta}$ is the predicted difference of $\Delta$, and $k_0$, $k_1$, $k_2$, $k_3$, $k_4$ and $k_5$ are parameters. The polynomial Eq. (3) serves the observation operators to eliminate the systematic difference between $\theta_s$ and $\theta_p$.

(4) Profile soil moisture could then be estimated by using the observation operators to rescale surface measurements.

$$\hat{\theta}_p = \theta_s - \hat{\Delta} \qquad (4)$$

where $\hat{\theta}_p$ is the predicted profile soil moisture.

### 2.2.2 Time series resolution effect

Three different temporal resolutions (hourly, daily, and weekly) were used to probe how time series resolution affects the performance of observation operators. To this end, three stations from SCAN with varying soil moisture values and surface-profile soil moisture dynamics were selected for analysis, including Shagbark Hills in Iowa, Perdido Riv Farm in Alabama, and Sevilleta in New Mexico. These stations were chosen because of the continuity and completeness of soil moisture datasets at both daily and hourly resolutions.

The testing procedure progressed as follows. First, hourly data within a given period were used to build observation operators. Surface moisture at daily (weekly) resolution during the same period was then as inputs into these observation operators to predict the corresponding daily (weekly) profile soil moisture. Next, observation operators derived from daily data were used to predict hourly (weekly) profile soil moisture in corresponding periods. Finally, observation operators derived from weekly data were used to predict hourly (daily) profile soil moisture in corresponding periods. Statistical metrics, including determination coefficient ($R^2$), root mean square error (RMSE) and mean bias error (MBE), were used to judge whether the developed observation operators were transferable between different temporal resolutions. The RMSE and MBE are defined as:

$$\text{RMSE} = \sqrt{\frac{1}{N} \sum_{i=1}^{N} (\theta_{obs,i} - \theta_{est,i})^2} \qquad (5)$$

and

$$\text{MBE} = \frac{1}{N} \sum_{i=1}^{N} (\theta_{obs,i} - \theta_{est,i}) \qquad (6)$$

where $\theta_{obs,i}$ and $\theta_{est,i}$ represent the observed and estimated profile soil moisture content, respectively, and $N$ is the number of soil moisture values in the corresponding time series.





### 2.2.3 Time series length effect

Three stations (Blue Creek, BC; Green River, GR; and Little Red Fox, LRF) in Utah with varying surface and profile soil moisture time series from 2010 to 2015, as well as continuous and complete datasets, were used to probe how time series length affects the performance of observation operators. Five different data lengths were chosen: a half-year growing season (Apr. 1 to Sept. 30; DL1); a half-year non-growing season (Oct. 1 to Mar. 31; DL2); one calendar year (DL3); two calendar years (DL4); and four calendar years (DL5). Four replicates were conducted for each data length. Specifically, soil moisture from the years of 2010, 2011, 2014, and 2015 was used to establish the observation operators for data lengths DL1, DL2 and DL3 (calibration), and data from 2012 and 2013 were used for validation. For data length DL4, soil moisture from four different two-year combinations (2010&2011, 2014&2015, 2010&2014 and 2011&2015) was used for calibration, and data from 2012 and 2013 were used for validation. As data length DL5 covers four years, four different four-year combinations (2010&2011&2012&2013, 2011&2012&2013&2014, 2012&2013&2014&2015, and 2010&2011&2014&2015) were used for calibration. The two years that were not included in each of the specified combinations were used for validation. Statistical metrics ($R^2$, RMSE, and MBE) were used to test the feasibility and robustness of observation operators built from soil moisture datasets that varied in the length of both calibration and validation periods. These analyses served to identify an appropriate data length for the construction of robust observation operators. Note that the surface measurements should also be ranked first during validation, after which they can act as inputs for the observation operators that derive ranked profile soil moisture.

### 2.2.4 Testing procedures in applications

The constructed observation operators were then applied to different climate regions to test how robust the predictions are for areas that vary in soil moisture content. Three primary climate regions in the continental USA were chosen (humid continental, humid subtropical, and semiarid) and three stations were selected for each climate region based on the continuity and completeness of soil moisture datasets. Detailed information regarding these nine stations is presented in Table 1.

### 2.3 Exponential filter

A popular analytical method, the exponential filter, the robustness of which has been validated in various climates (Wagner et al., 1999; Ceballos et al., 2005; Albergel et al., 2008; Brocca et al., 2011; Ford et al., 2014; Gao et al., 2014; Peterson et al., 2016), served as a reference method to judge the performance of observation operators in different climate regions.

According to Wagner et al. (1999), a soil profile can be divided into the surface layer and a second (subsurface) layer. The exponential filter was introduced to predict second-layer soil moisture ($w_2$) from surface measurements ($w_1$). This method assumes that the water flux between surface and subsurface layers is proportional to the difference in soil moisture between these two layers. The surface and subsurface soil moisture can be linked as follows:





$$L_2 \frac{dw_2(t)}{dt} = C\left[w_1(t) - w_2(t)\right] \tag{7}$$

Assuming $C$ is constant and $T=L/C$, the solution of Eq. (7), is derived by integration as:

$$w_2(t) = \frac{1}{T}\int_{-\infty}^{t} w_g(\tau)\exp(-\frac{t-\tau}{T})d\tau \tag{8}$$

where $L_2$ is the depth of the second layer, $C$ is the area-representative pseudo-diffusivity constant, and $T$ represents the

characteristic time length. Generally, $T$ is considered to be an integrative parameter that explains all the hydrologic, pedologic and ecological processes that influence soil moisture variations with depth (Wagner et al., 1999; Albergel et al., 2008). To simplify the computation, Albergel et al. (2008) gave the recursive formulation of Eq. (8) as follows:

$$\mathrm{SWI}_{2(t)} = \mathrm{SWI}_{2(t-1)} + K_t[\mathrm{ms}_{(t)} - \mathrm{SWI}_{2(t-1)}] \tag{9}$$

and

$$K_t = \frac{K_{t-1}}{K_{t-1} + \exp(\frac{\Delta t}{T})} \tag{10}$$

where ms represents soil moisture at the surface layer, and $\mathrm{SWI}_2$ represent the soil water index (SWI) of the second layer; $K_t$ is the gain of the exponential filter. The dimensionless SWI represents the scaled soil moisture content, which ranges from 0 to 1 based on the minimum and maximum values of each time series.

It is important to note that predicted second-layer soil moisture ($\mathrm{SWI}_2$) *via* Eq. (9) provides scaled values. To compare these values with results from observation operators, $\mathrm{SWI}_2$ must be rescaled by using the maximum ($w_{2,\max}$) and minimum ($w_{2,\min}$) values of the corresponding original time series of the second layer as follows:

$$\hat{w}_2(t) = \mathrm{SWI}_{2(t)}(w_{2,\max} - w_{2,\min}) + w_{2,\min} \tag{11}$$

where $\hat{w}_2(t)$ is the rescaled value of predicted soil moisture at the second layer, $\mathrm{m}^3\,\mathrm{m}^{-3}$. The rescaled value $\hat{w}_2(t)$ and the

surface measurements $w_1(t)$ can then be coupled to obtain the profile soil moisture as:

$$\hat{w}_p(t) = \frac{\hat{w}_2(t)\times L_2 + w_1(t)\times L_1}{L_2 + L_1} \tag{12}$$

where $\hat{w}_p(t)$ is the predicted profile soil moisture at time $t$, and $L_1$ is the depth of surface layer.





## 3 Results & discussion

### 3.1 The effect of time series resolution and length

The surface and profile soil moisture time series at different temporal resolutions for the Shagbark Hills, Perdido Riv Farms and Sevilleta stations are used to probe the effect of data resolution on the performance of observation operators. The

statistical metrics, i.e., RMSE, $R^2$ and MBE, for the application of observation operators built from a dataset of a certain resolution to another dataset with a different resolution are presented in Table 2. Generally, observation operators that had been calibrated using datasets of a different resolution performed well; there was little variation between the three statistical metrics from all stations when the operators were transferred from finer (coarser) to coarser (finer) resolutions. Overall, the temporal resolution of soil moisture time series had a relatively weak effect on the performance of observation operators, and

hence can be ignored during the construction of observation operators. In this study, observation operators were built from soil moisture data at a daily resolution in order to reach a compromise between computing efficiency and operator robustness.

The statistics of data length effects on the performance of observation operators in both calibration and validation periods are shown in Figure 3. In general, observation operators performed better in calibration than in validation periods, especially for

half-year durations (DL1 and DL2). Furthermore, these metrics behaved differently between calibration and validation periods with the increase of data lengths. In calibrations, lower values of RMSE were observed at half-year durations (DL1 and DL2), peaked at one-year duration (DL3) and then decreased slightly for all of the three stations; and the values of $R^2$ in DL1 and DL2 were clearly higher than in longer durations. In contrast, in validations, longer durations (DL3, DL4 and DL5) generally showed lower values in RMSE and MBE and higher values in $R^2$, indicating higher prediction accuracy in longer

duration. Moreover, longer data durations (DL3, DL4 and DL5) generally showed clearly lower uncertainty in either calibrations or validations, represented by lower error bars in Figure 3, than the other two shorter durations (DL1 and DL2). These results indicated that the observation operators became more representative, producing more robust predictions of profile soil moisture in longer data lengths. Compared to calibration periods, the performance in validation periods can serve as a better reference for judging data length effects because independent datasets are employed. Although the DL3, DL4 and

DL5 showed almost equally good mean values of RMSE, $R^2$ and MBE in validation periods, the DL4 overall showed the lowest uncertainty for all metrics in either calibration or validation period. Therefore, the observation operators in this study were built based on a data length of two years.

### 3.2 Application in various climates

In this section, we applied the method of observation operators built by the CDF matching method under nine stations

distributed across three primary climate regions in the continental USA. Here, we first used correlation analysis to





characterize the couple strength of surface and profile soil moisture. Then the feasibility and robustness of observation operators was tested by employing the exponential filter as a reference method.

### 3.2.1 Correlation analysis

Lagged cross correlation analysis was used to characterize surface-profile soil moisture relationships and autocorrelations of each of surface and profile soil moisture were performed as well under various climates. This is because subsurface soil moisture shows a delayed response to atmospheric variables (e.g. precipitation and evapotranspiration) and cross correlation analysis is able to characterize lagged relations (Georgakakos et al., 1995; Mahmood et al., 2012; Ford et al., 2014). As shown in Figure 4, the cross correlation coefficient peaked at no lag or one-day lag and the stations of Perdido Riv Farms

and Willow Wells showed clear periodicity. It is clear that the maximum correlation coefficient varied with climates; the humid subtropical climate had the highest correlation coefficient of 0.92±0.01 (mean ± one standard deviation), followed by humid continental climate of 0.75±0.13 and semiarid climate of 0.59±0.19. This means that surface and profile soil moisture under humid subtropical climate are better coupled than the other climates and the profile soil moisture is expected to be better predicted under this climate *via* depth scaling methods. The higher cross correlation coefficients in humid subtropical

can be ascribed into the highly similar autocorrelation patterns of surface and profile soil moisture (Figure 4). However, autocorrelations of surface and profile soil moisture varied clearly in magnitude and behaviour in station of Molly Caren under humid continental and stations of Holden and Sevilleta under semiarid, as can be responsible to the relatively low cross correlations there.

### 3.2.2 Testing of observation operators

The first step of the exponential filter method is the calculation of the optimal value for the $T$ parameter ($T_{opt}$), which is obtained by maximizing the Nash-Sutcliffe coefficient (NSC; Nash and Sutcliffe, 1970); the $T$ value at which NSC peaks is regarded as the $T_{opt}$ for a given dataset (Wagner et al., 1999; Albergel et al., 2008). The relationships between the $T$ parameter and NSC at various stations in calibration period are shown in Figure 5. The $T_{opt}$ varied from 1 to 23 days for

different stations, except for the Holden station, which had negative NSC values over the whole range of $T$ values under consideration (0-50 days) and did not even peak when $T$ was set to 100 days (data not shown here). In-depth analysis showed that the $T_{opt}$ value was highly dependent on the type of climate. The average $T_{opt}$ values for the humid continental, humid subtropical and semiarid climates were 14.0, 1.3 and 17.5 days, respectively. This is partly consistent with the findings of Albergel et al. (2008), who also found that climate impacts $T_{opt}$, yet reported a relatively weak effect. According to Albergel

et al. (2008), the $T_{opt}$ reflects the response of second-layer soil moisture ($w_2$) to surface moisture ($w_1$), with a higher $T_{opt}$ representing a slower response. In the humid subtropical climate, a relatively high soil moisture content over the whole profile (see Figure 9) reflects large soil hydraulic conductivity and a relatively fast response of $w_2$ to $w_1$. In the semiarid



climate, however, both surface and subsurface soil moistures are relatively low, which results in low soil hydraulic conductivity and a slower second-layer response to surface infiltration.

Prior to the testing of observation operators, each soil moisture time series was checked to avoid any outlier. Soil moisture
datasets from 2014 and 2015 were used as calibration data for both methods at all stations except for Perdido Riv Farms and Willow Wells, where datasets from 2013 and 2014 and from 2012 and 2013 were employed, respectively. Data from either 2013 (2016 for Perdio Riv Farms, and 2010 for Willow Wells) were used for validation depending on the completeness of dataset. Figures 6 to 8 show the predicted profile soil moisture time series from observation operators and exponential filters in humid continental, humid subtropical and semiarid climates, respectively. Generally, the observation operators reproduced
reliably the profile soil moisture for the majority of stations in both calibration and validation periods, and performed overall equally well with the exponential filter method. This can be attributed to the perfect adjusting of the cumulative distribution frequencies of surface moisture and profile data by observation operators (the left graphs in Figure 6-8). However, some clear deviations between measured and predicted values were observed in Molly Caren Station during calibration period and Holden and Willow Wells stations during validation period. These significant mismatches can be attributed to the relatively
poor coupling of surface and profile soil moisture (Figure 4). Statistical metrics of the predicted results for all three climates are shown in Figure 9. Similar RMSE and $R^2$ values were observed between the observation operators and exponential filter methods during either calibration or validation period, which agrees with the visual observations in Figure 6-8. However, for the metric of MBE, zero values for observation operators were observed in all stations but clear non-zero values for the exponential filter method during calibration period. In validation period, non-zero values of MBE were also observed for
observation operators. Overall, the results indicated that observation operators built from a soil moisture dataset with a length of two years were feasible in scaling surface soil moisture to obtain profile predictions.

We further analysed the effects of climates on the performance of both observation operators and the exponential filter method according to the statistics in Figure 9. The ANOVA results indicated that the climate region influenced to different
extents the predictions of profile soil moisture for both methods. A semiarid climate showed significantly ($P<0.05$) lower RMSE than the other climates in calibration period independent of methods. However, this did not mean that the semiarid climate had better prediction accuracy than the other climates because it also showed significantly ($P<0.05$) lower $R^2$ in both calibration and validation periods. From Figure 8, it can be seen that mismatches existed between measured and predicted values through the whole study period independent of methods, and the relatively lower RMSE can be ascribed into the
lower soil moisture contents there. Combining the findings in section 3.2.1 that clearly lower maximum cross correlations under semiarid climate (Figure 4), it can be gained that a relatively poor coupling between surface and subsurface soil moisture exists in semiarid climates. Furthermore, the humid subtropical climate showed clearly lower RMSE and meanwhile the highest $R^2$ ($>0.90$) compared to the other climates, indicating relatively good coupling of surface and



subsurface soil moisture there. A possible explanation is that the humid subtropical has generally high soil moisture content (Figure 7) and possible high soil hydraulic properties because of silt loam soils there (Silver City in Table 1), which could strengthen the connections of water flow between surface and subsurface soils.

**3.3 Discussion**

Here we introduced a simple and feasible statistical method, the CDF matching method, in predicting profile soil moisture from surface observations. The application of this method in various climates demonstrates it can be applied to a variety of environments if there is long enough soil moisture time series (two years). On the one hand, the advantage of this method over other statistical methods such as linear and non-linear regressions (Bono and Alvarez, 2012) and time stability analysis

(Hu et al., 2014; Gao et al., 2015) is that only surface (5 cm) soil moisture is needed as the input. And its advantage relative to data assimilation and analytical models is that no clear assumption is needed in addition to its simplicity. On the other hand, the primary disadvantages of this method can be summarized into two aspects. First, outliers in soil moisture data can affect greatly the robustness of observation operators built by the CDF matching method, and hence outliers should be carefully kicked out before building observation operators. Second, the deviations between predicted and measured soil

moisture cannot be interpreted from the perspective of physically based grounds.

Uncertainty is generally one of the main issues when statistical methods are applied. In calibration, the primary uncertainty of the method here can be attributed into the fitting curves (five-order observation operators). Generally, the fitting curve cannot completely match the relationship between surface moisture content ($\theta_s$) and the difference ($\Delta$) between surface and

profile soil moisture. In validation, the relationship between $\theta_s$ and $\Delta$ could deviate to some extent with respect to that in calibration, and this deviation is expected to increase prediction errors. Moreover, it is noteworthy that the method here cannot replace data assimilation and analytical models in depth scaling of soil moisture because it completely does not work in ungauged regions. However, this method could be a good choice to obtain profile soil moisture if the target site has long enough soil moisture observations even with coarse temporal resolutions.

**4. Conclusions**

In this study, we tested the feasibility and robustness of observation operators built by Cumulative Distribution Frequency (CDF) matching for the depth scaling of soil moisture from surface to profile by using datasets of multiple stations in the Soil and Climate Analysis Network (SCAN). The time series resolution (hourly, daily and weekly) had negligible effects on the performance of observation operators and daily resolution provided a good compromise between computing efficiency

and operator robustness. In contrast, time series length was shown to significantly affect the prediction accuracy of observation operators. Our analyses showed that soil moisture data from a two-year interval produced optimal predictions,





and can be used as the standard dataset from which to build observation operators. By using the popular exponential filter method as a reference approach we were able to show that the application of observation operators to three primary climates (humid continental, humid subtropical and semiarid) in the continental USA can reliably predict profile (0-100 cm) soil moisture from only surface (5 cm) observations for the majority of stations. The CDF matching method described here may

be applicable to depth scaling of soil moisture under a variety of environments if the target site has long enough (two years) soil moisture observations even with coarse temporal resolutions.

*Acknowledgements*. We thank Min Yang, Liuyang Yu and Mengyan Jiao for their helps in data downlaod and processing. This work was jointly supported by the National Natural Science Foundation of China (No. 41401315, 41571506, 51579212),

the National Key Research and Development Plan (No. 2016YFC0400204), the '111' Project (No. B12007) and the Integrative Science-Technology Innovation Engineering Project of Shaanxi (No. 2016KTZDNY-01-03).

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



**Table 1.** The geographical information of SCAN sites used in this study.

| Site name | Site ID | State* | Climate region** | Latitude (degree) | Longitude (degree) | Soil texture*** | | | | | Elevation (m) |
|---|---|---|---|---|---|---|---|---|---|---|---|
| | | | | | | 5 cm | 10 cm | 20 cm | 50 cm | 100 cm | |
| Blue Creek | 2135 | UT | Mediterranean continental | 41.93 N | 112.43 W | SIL | SIL | SIL | SICL | SIL | 1582 |
| Centralia Lake | 2094 | KS | Humid continental | 39.07 N | 96.17 W | SIC | SIC | SIC | SIC | SICL | 397 |
| Green River | 2131 | UT | Semiarid | 39.02 N | 110.17 W | FSL | FSL | FSL | L | FSL | 1252 |
| Holden | 2127 | UT | Semiarid | 39.20 N | 112.40 W | FSL | FSL | FSL | L | SL | 1445 |
| McCracken Mesa | 2140 | UT | Highland | 37.45 N | 109.33 W | VFSL | VFSL | VFSL | L | VFSL | 1621 |
| Molly Caren | 2014 | OH | Humid continental | 39.95 N | 83.45 W | SIL | SIL | SIL | C | L | 323 |
| Perdido Riv Farms | 2181 | AL | Humid subtropical | 31.12 N | 87.33 W | / | / | / | / | / | 91 |
| Sevilleta | 2171 | NM | Semiarid | 34.35 N | 106.68 W | / | / | / | / | / | 1595 |
| Shagbark Hills | 2068 | IA | Humid continental | 42.43 N | 95.77 W | SICL | SICL | SICL | SICL | SICL | 427 |
| Silver City | 2086 | MS | Humid subtropical | 33.08 N | 90.52 W | SIL | SIL | SIL | SICL | SICL | 35 |
| Willow Wells | 2108 | NM | Semiarid | 33.53 N | 103.63 W | FS | FS | FS | SCL | COSL | 1383 |
| Youmans Farm | 2038 | SC | Humid subtropical | 32.67 N | 81.20 W | / | / | / | / | / | 23 |

*Abbreviations in state: AL.: Alabama, IA: Iowa, KS: Kansas, MS: Mississippi, NM: New Mexico, OH: Ohio, SC: South Carolina, UT: Utah

**Köppen climate classification

***Abbreviations in soil texture: C: clay, CL: clay loam, COSL: coarse sandy loam, FS: fine sand, FSL: fine sandy loam, L: loam, LCOS:

5   loamy coarse sand, LFS: loamy fine sand, S: sand; SCL: sandy clay loam, SIC: silty clay, SICL: silty clay loam, SIL.: silt loam, VFSL: very
fine sandy loam.





**Table 2.** Statistics of the performance of observation operators built by dataset of one resolution and their applications in datasets with other resolutions. The statistical metrics were root mean square error (RMSE, $m^3\ m^{-3}$), $R^2$ and mean bias error (MBE, , $m^3\ m^{-3}$).

| Statistics | Shagbark Hills | | | Perdido Riv Farms | | | Sevilleta | | |
|---|---|---|---|---|---|---|---|---|---|
| | RMSE | $R^2$ | MBE | RMSE | $R^2$ | MBE | RMSE | $R^2$ | MBE |
| Calibrated by hourly data | 2.91E-2 | 0.774 | 0.000 | 1.07E-2 | 0.878 | 0.000 | 1.04E-2 | 0.445 | 0.001 |
| Applied to daily data | 2.85E-2 | 0.779 | 0.000 | 9.04E-3 | 0.905 | 0.000 | 8.95E-3 | 0.547 | 0.000 |
| Applied to weekly data | 2.91E-2 | 0.775 | 0.000 | 9.32E-3 | 0.905 | 0.000 | 9.67E-3 | 0.472 | 0.000 |
| Calibrated by daily data | 2.86E-2 | 0.778 | 0.000 | 9.11E-3 | 0.905 | 0.000 | 9.15E-3 | 0.544 | 0.000 |
| Applied to hourly data | 2.93E-2 | 0.772 | 0.000 | 1.08E-2 | 0.877 | 0.000 | 1.54E-2 | 0.556 | 0.001 |
| Applied to in weekly data | 2.92E-2 | 0.774 | -0.001 | 9.38E-3 | 0.905 | 0.000 | 9.96E-3 | 0.468 | 0.000 |
| Calibrated by weekly data | 2.94E-2 | 0.772 | 0.000 | 9.34E-3 | 0.906 | 0.000 | 9.89E-3 | 0.463 | 0.000 |
| Applied to hourly data | 2.98E-2 | 0.768 | 0.001 | 1.09E-2 | 0.877 | 0.000 | 1.84E-2 | 0.503 | 0.001 |
| Applied to daily data | 2.92E-2 | 0.773 | 0.001 | 9.12E-3 | 0.905 | 0.000 | 9.17E-3 | 0.539 | 0.000 |





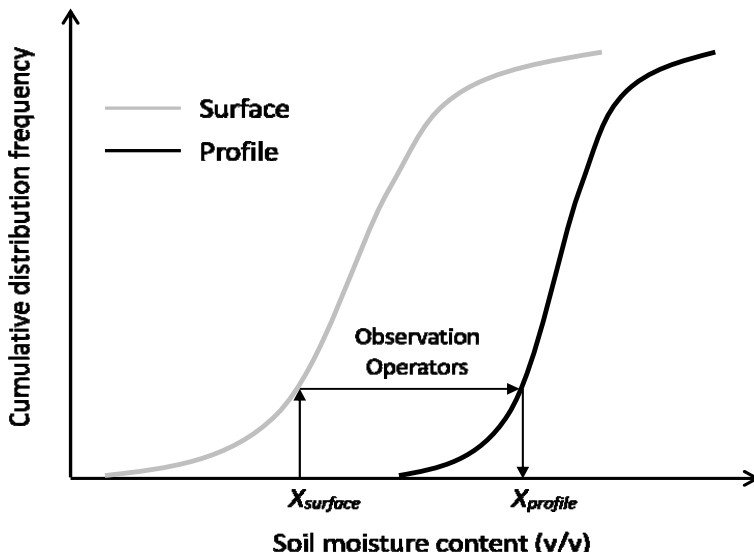

**Figure 1.** A graph shows how the cumulative distribution frequency (CDF) of surface soil moisture is adjusted into that of
profile soil moisture by observation operators.



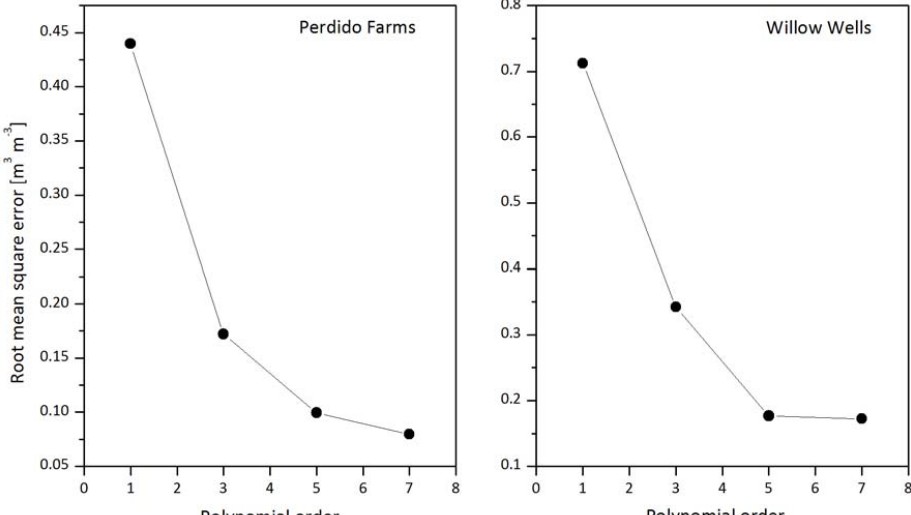

**Figure 2.** Root mean square error (RMSE) changes with the increasing polynomial order by using the cumulative distribution function (CDF) matching method.



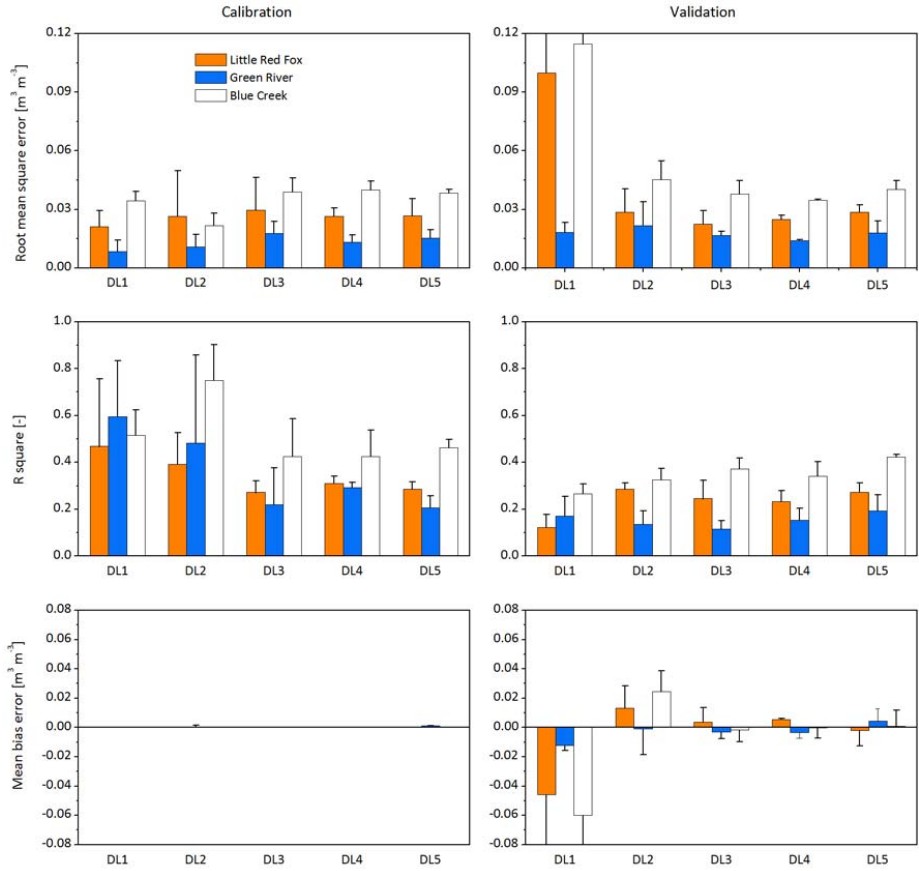

**Figure 3.** Statistics of the root mean square error (RMSE), $R^2$ and mean bias error (MBE) for data length of soil
5   moisture time series on the performance of observation operators in both calibration and validation periods. The error bar
represents one standard deviation.





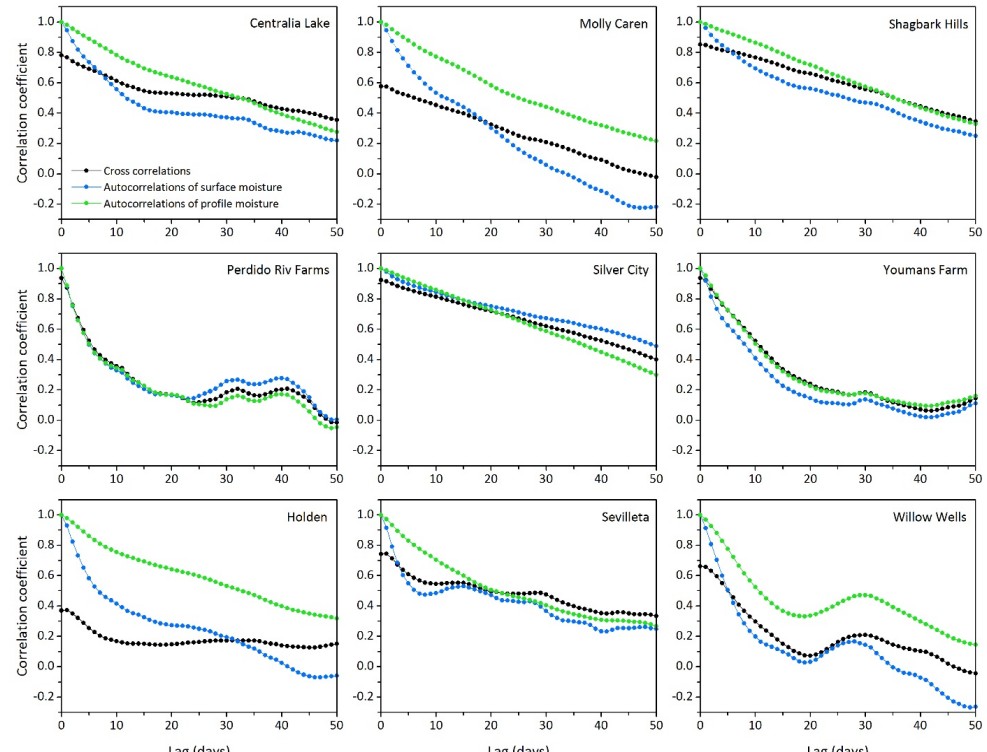

**Figure 4.** Cross correlations and autocorrelations of surface and profile soil moisture in the nine stations under various
5   climates.





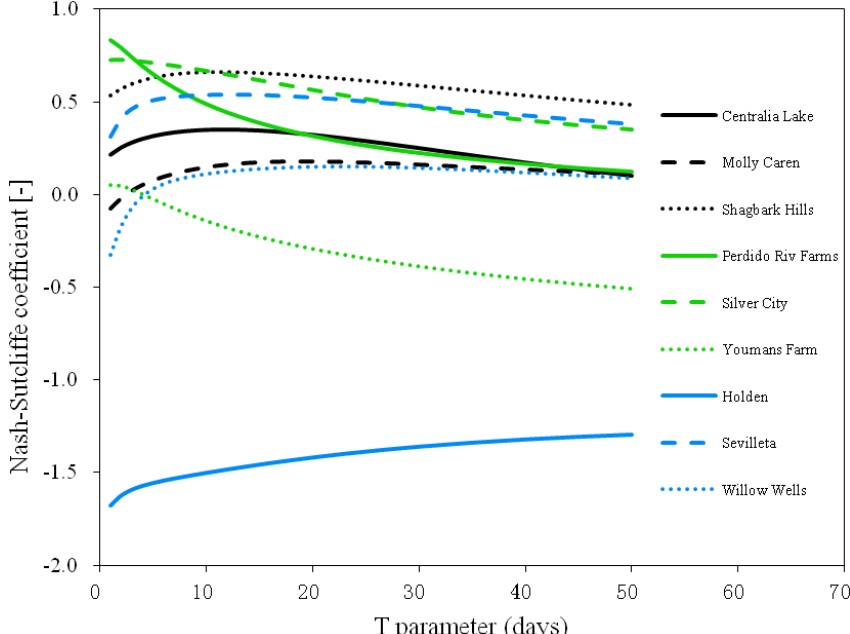

**Figure 5.** A graph shows how the cumulative distribution frequency (CDF) of surface soil moisture is adjusted into that of profile soil moisture by observation operators.





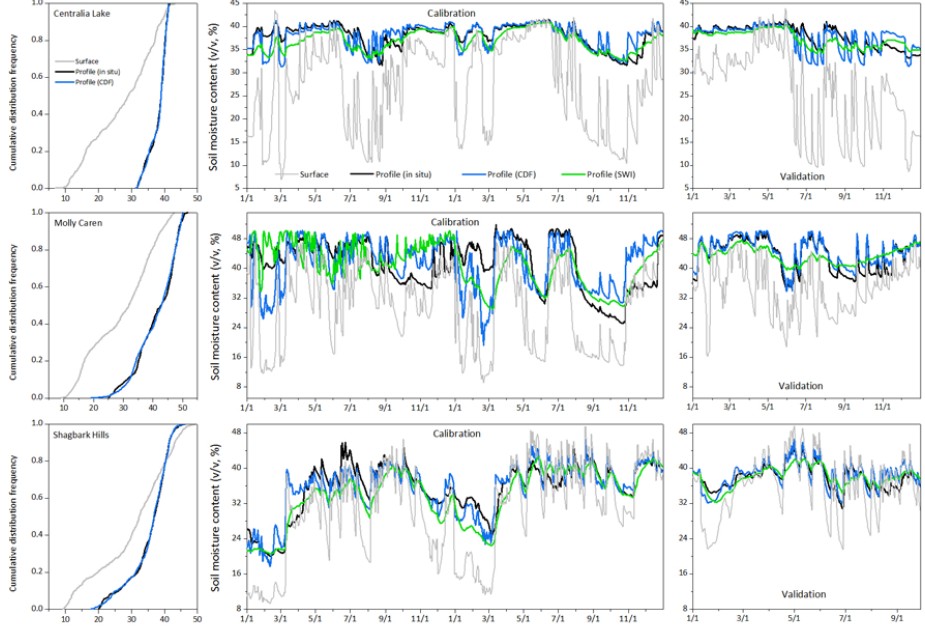

5 **Figure 6.** The predicted profile soil moisture by cumulative distribution frequency (CDF) matching (Profile (CDF)) and exponential filter (Profile (SWI)) during calibration and validation periods for the three sites in humid continental climate. The three graphs in the left shows the CDFs of surface and measured and predicted profile soil moisture.





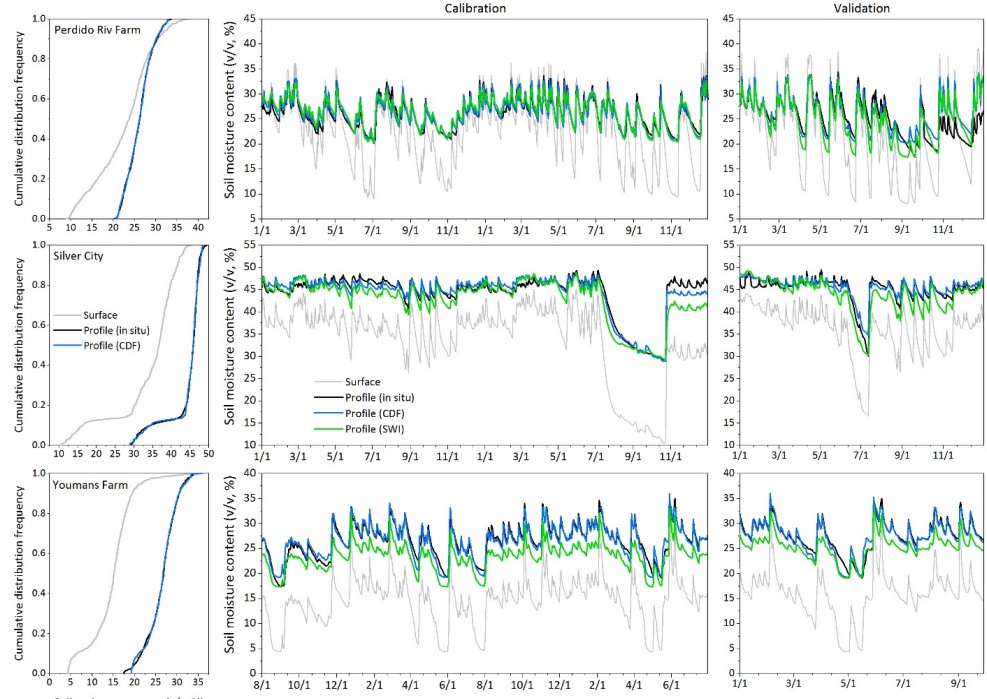

5 **Figure 7.** The predicted profile soil moisture by cumulative distribution frequency (CDF) matching (Profile (CDF)) and exponential filter (Profile (SWI)) during calibration and validation periods for the three sites in humid subtropical climate. The three graphs in the left shows the CDFs of surface and measured and predicted profile soil moisture.





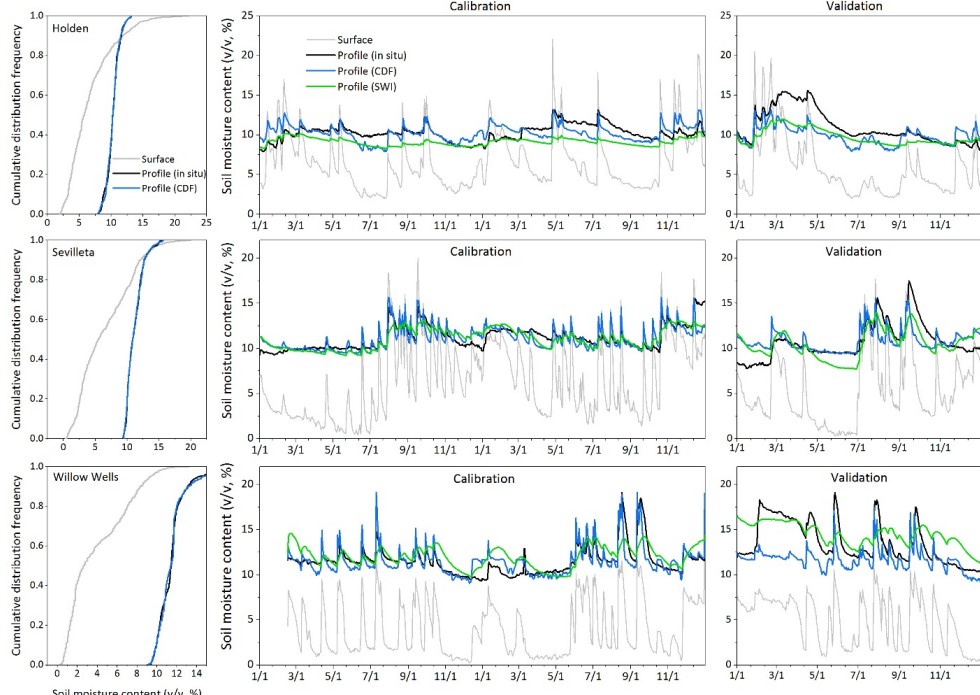

5 **Figure 8.** The predicted profile soil moisture by cumulative distribution frequency (CDF) matching (Profile (CDF)) and exponential filter (Profile (SWI)) during calibration and validation periods for the three sites in semiarid climate. The three graphs in the left shows the CDFs of surface and measured and predicted profile soil moisture.




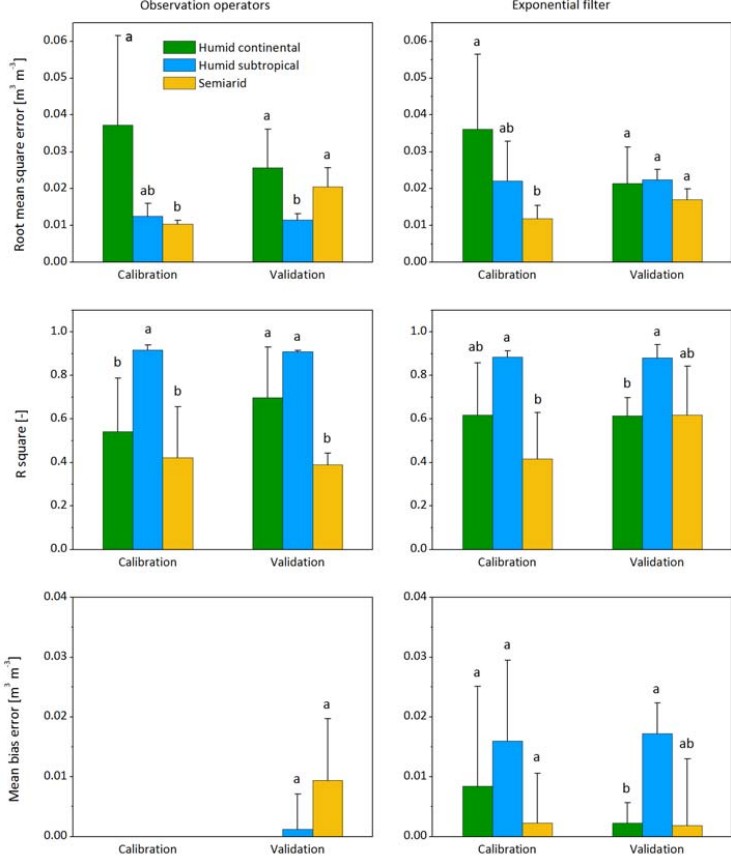

**Figure 9.** Statistics of the root mean square error (RMSE), $R^2$ and mean bias error (MBE) for the performance of observation
operators and exponential filter methods in the different sites in humid continental, humid subtropical and semiarid climates.
Different lowercase letters above bars indicate significant ($P<0.05$) differences between climates in either calibration or
validation period.