# Peer review of "Depth scaling of soil moisture content from surface to profile: multistation testing of observation operators"

_Hydrology and Earth System Sciences, 2017_

## Referee Comment (RC1) · Anonymous Referee #1 · 17 Jul 2017

It's my pleasure to review hess-2017-292 "Depth scaling of soil moisture content from surface to profile: multi-station testing of observation operators" by Gao et al. The authors try to use Cumulative Distribution Frequency (CDF) matching method to build the observation operators and adopt this method to predict profile soil moisture from surface measurements. This is a re-submitted manuscript with previous ID "hess-2016-617". The authors do not provide a response to previous reviewer's comments to indicate what has been changed in comparison to previous version, and I find that the results presented in the two versions are significantly different while the data and method are identical. In addition, I'm curious about the transferability of the proposed method, for example, how can the authors apply their method to satellite products? According

to these, I suggest to reject this paper. My concerns are as follows.

Major Concern

1. The results presented in current version are significantly different from the previous version while the adopted data and method are identical, for example, Table 2 vs. Table 2 in hess-2016-617, Figure 3 vs. Figure 5 in hess-2016-617, why?

2. It's suggested to provide a response to previous reviewer's comments to indicate what has been changed.

3. A fifth-order polynomial fit is adopted, but I do not see any fit parameters for the selected stations. In addition, I'm wonder the transferability of these fit parameters, for example, can these fit parameters applicable for the similar climate condition without further calibration?

4. The authors mention the prediction of profile soil moisture from satellite based surface measurements in the Introduction part, I'm thus curious about how can the authors apply their method to satellite products?

5. In the Introduction part, the authors also argue that "continuous and accurate measurement of profile soil moisture, however, is difficult because of expensive field measurements", but they also indicate in the Abstract that "the findings here have the potential to be applied in profile soil moisture prediction from surface measurements at a range of environments if the target site has long enough (two years) soil moisture observations even with coarse temporal resolutions", then I'm wonder how their methods address the drawback of in situ profile soil moisture measurements, since the methods depend on the calibration that also needs the profile soil moisture measurements.

6. The Title of the manuscript is confusing, for example, what do you mean by "depth scaling", and what's the meaning of "observation operators".

Minor Concern

1. Page 2, Line 4, can the cosmic-ray probes measure surface soil moisture directly?

2. Page 3, Line 29, "A total of 12 stations were chosen for analyses according to the objectives of this study". This sentence is not clear. I still do not understand why the authors only select 12 stations out of the more than 200 SCAN stations. In the previous version, the authors mentioned that 31 stations were selected, why the numbers are changed?

3. Page 4, equation (1), I do not understand why the authors use this equation to calculate profile soil moisture, please provide corresponding reference.

4. Page 6, Line 6, "Specifically, soil moisture from the years of 2010, 2011, 2014, and 2015 was used to establish the observation operators for data lengths DL1, DL2 and DL3 (calibration), and data from 2012 and 2013 were used for validation", what is the reason for such a division, for example, why don't you use 2014 and 2015 for validation, and other years for calibration? What can be the impact?

―――――――――――――――――――――

---

## Short Comment (SC1) · 18 Jul 2017

Please see the supplement for our responses to the comments of previous reviewers for hess-2016-617.

Please also note the supplement to this comment:
https://www.hydrol-earth-syst-sci-discuss.net/hess-2017-292/hess-2017-292-SC1-supplement.zip

---

## Author Comment (AC1) · 18 Jul 2017

**Responses to the comments of Anonymous Referee #1**

It's my pleasure to review hess-2017-292 "Depth scaling of soil moisture content from surface to profile: multi-station testing of observation operators" by Gao et al. The authors try to use Cumulative Distribution Frequency (CDF) matching method to build the observation operators and adopt this method to predict profile soil moisture from surface measurements. This is a re-submitted manuscript with previous ID "hess-2016-617". The authors do not provide a response to previous reviewer's comments to indicate what has been changed in comparison to previous version, and I find that the results presented in the two versions are significantly different while the data and method are identical. In addition, I'm curious about the transferability of the proposed method, for example, how can the authors apply their method to satellite products? According to these, I suggest to reject this paper. My concerns are as follows.

>> Thanks a lot for your comments. Yes, this is a resubmitted manuscript based on the previous rejected paper hess-2016-617. We had addressed every single comment of the two reviewers of the paper hess-2016-617 whereas failed to upload the responses before the due time because of a number of reasons. But we emailed our responses to the two reviewers (Prof. Wolfgang Wagner and Prof. Na Li) before the resubmission and they generally agree with our responses. The resubmitted paper hess-2017-292 is a thoroughly revised manuscript according to the reviewer's comments. We have made recalculations for all stations in question because there is something wrong with the original procedure of the CDF matching. We have also redrawn all figures. This is why the results in the hess-2017-292 are clearly different with the former one. During the resubmission procedure, we were not sure whether we should upload the responses and thus we only upload the revised paper. And this time we have added the responses to the comments of previous reviewers as a supplement by posing a short comment. In this manner, the responses can be more visible to other referees and readers.

For the transferability, every method, including the data assimilation and analytical methods, in fact need prior soil moisture data to improve and test its robustness. In this paper, we mean that if the observation operators (polynomial) derived by the CDF matching is tested robust, we can use remote sensing soil moisture (generally shallower than 5 cm) as inputs to predict profile soil moisture *via* these simple operators.

Major Concern

1. The results presented in current version are significantly different from the previous version while the adopted data and method are identical, for example, Table 2 vs. Table 2 in hess-2016-617, Figure 3 vs. Figure 5 in hess-2016-617, why?

>> In fact, there was some wrong with the procedure of the CDF matching method in the hess-2016-617 and thus produced unbelievable results. According to the comments of Prof. Wagner, we corrected this procedure, made recalculations and redrew these figures. Therefore, the Table 2, Figures and also texts in hess-2017-292 significantly different with the former paper.

2. It's suggested to provide a response to previous reviewer's comments to indicate what has been changed.

>> We agree. We have uploaded the responses to previous reviewer's comments as a supplement

by posing a short comment. In this manner, the responses can be more visible to other referees and readers.

3. A fifth-order polynomial fit is adopted, but I do not see any fit parameters for the selected stations. In addition, I'm wonder the transferability of these fit parameters, for example, can these fit parameters applicable for the similar climate condition without further calibration?
>> Thanks for the comments. We did not indicate these parameters in the paper and did not test the spatial transferability of observation operators in a given climate. This is interesting, and we would do this work and show the results if we have a chance to revise our paper.

4. The authors mention the prediction of profile soil moisture from satellite based surface measurements in the Introduction part, I'm thus curious about how can the authors apply their method to satellite products?
>> In this paper, we mean that if the observation operators (polynomial) derived by the CDF matching is tested robust, we can use remote sensing soil moisture (shallower than 5 cm) as inputs to predict profile soil moisture *via* these simple operators.

5. In the Introduction part, the authors also argue that "continuous and accurate measurement of profile soil moisture, however, is difficult because of expensive field measurements", but they also indicate in the Abstract that "the findings here have the potential to be applied in profile soil moisture prediction from surface measurements at a range of environments if the target site has long enough (two years) soil moisture observations even with coarse temporal resolutions", then I'm wonder how their methods address the drawback of in situ profile soil moisture measurements, since the methods depend on the calibration that also needs the profile soil moisture measurements.
>> To our knowledge, every method, including the data assimilation and analytical methods, in fact need prior surface and profile soil moisture data to calibrate the parameters and validate its feasibility and robustness. Here we mean that if robust observation operators are built we only need surface measurements to obtain profile soil moisture.

6. The Title of the manuscript is confusing, for example, what do you mean by "depth scaling", and what's the meaning of "observation operators".
>> Depth scaling means scale surface observations to profile soil moisture. Observation operators is first introduced by Reichle and Koster (2014) and it denotes the polynomial built by the CDF matching method.

*Reichle, R.H., and Koster, R.D.: Bias reduction in short records of satellite soil moisture, Geophys. Res. Lett., 31 (19), 2004.*

Minor Concern
1. Page 2, Line 4, can the cosmic-ray probes measure surface soil moisture directly?
>> We have not used the cosmic-ray probe to measure soil moisture. We got this knowledge from the literature, for example, Peterson et al. (2016) where they defined surface soil as the top 20 cm.

*Peterson, A.M., Helgason, W.D., and Ireson, A.M.: Estimating field-scale root zone soil moisture using the cosmic-ray neutron probe. Hydrol. Earth Syst. Sci., 20, 1373-1385, 2016.*

2. Page 3, Line 29, "A total of 12 stations were chosen for analyses according to the objectives of this study". This sentence is not clear. I still do not understand why the authors only select 12 stations out of the more than 200 SCAN stations. In the previous version, the authors mentioned that 31 stations were selected, why the numbers are changed?

>> In fact, we also used 12 stations in the hess-2016-617. The text that 31 stations were used was a careless mistake. Please also see our explanations in the responses to previous reviewer's comments in the supplements. In this paper, we used three stations as three replicates for each of three climate regions and three other stations to test the effects of data lengths on the performance of observation operators. We only used 12 stations but not all of the SCAN stations for two primary reasons. On the one hand, we argue that three replicates are generally enough to test the feasibility of the observation operators. On the other hand, a lot of SCAN stations lacked considerable data at one or several depths especially in the humid continental and humid subtropical climates. We selected these stations because they have only a few miss in the data.

3. Page 4, equation (1), I do not understand why the authors use this equation to calculate profile soil moisture, please provide corresponding reference.

>> This equation considered the weight of soil thickness since the thickness of different depth intervals was not identical. It can be referred to Hu and Si (2014).

*Hu, W., and Si, B.C.: Can soil water measurements at a certain depth be used to estimate mean soil water content of a soil profile at a point or at a hillslope scale. J. Hydrol., 516, 67-75, 2014.*

4. Page 6, Line 6, "Specifically, soil moisture from the years of 2010, 2011, 2014, and 2015 was used to establish the observation operators for data lengths DL1, DL2 and DL3 (calibration), and data from 2012 and 2013 were used for validation", what is the reason for such a division, for example, why don't you use 2014 and 2015 for validation, and other years for calibration? What can be the impact?

>> Initially, the maximum data length was set as two years when analyze the effects of data length on observation operators. The data in the 2010 and 2011 were used for calibration and that in 2012 and 2013 for validation. Afterwards, we extended the maximum data length to four years and the data in 2014 and 2015 were used. In our opinion, this arrangement of data for calibration and validation has little impact on the results because there is no scientific rule to our knowledge that calibration should use earlier data than validation.

---

## Referee Comment (RC2) · Y. Zeng (Referee) · 21 Jul 2017

The authors developed a CDF matching-based approach to estimate the profile soil moisture. The preliminary application of the approach over different climate zones (e.g. with a certain number of in situ stations) shows the promising results. I have only two main concern as follows:

Major concern: 1. The author indicated the validation and calibration periods specifically for the current scaling approach. On the other hand, it is not clear why the certain year was selected for validation, why the certain year was selected for calibration. Please, the author clarifies, for example, why 2012, 2013 selected for validation,

instead of 2014, 2015? I'm also curious how the selection of different cal/val periods will affect the results.

2. For figure 5, you can find negative NSC. There is no further discussion in the paper to explain why. Does it imply a limitation for applying the current developed CDF matching-based approach?

Minor comments: See supplement.

Please also note the supplement to this comment:
https://www.hydrol-earth-syst-sci-discuss.net/hess-2017-292/hess-2017-292-RC2-supplement.pdf

---

## Short Comment (SC2) · 28 Jul 2017

The manuscript presents a statistic approach to derive Root-zone soil moisture (0-100 cm) from near-surface soil moisture measurements (5cm). This is an important and current topic since remote sensing techniques devoted to measure soil moisture are only sensitive to the top 5 cm soil moisture and most of applications (agriculture, meteorology, hydrology) are interested in deeper soil moisture estimates. The proposed methodology consists in using a CDF-matching procedure applied to the 5 cm soil moisture data to predict 0-100 soil moisture dynamic. To my knowledge, this has not been tested in previous studies. Authors address the problem of three issues: the calibration period-length, the temporal sample, and the climatic effect on the surface/deep soil moisture link. Dataset comes from the US SCAN network. Although the subject is interesting and deserves publication in HESS, the paper presents some deficiencies. For this reason, I would recommend major revisions prior to potential publication in HESS.

General comments:

The main problem of the proposed methodology rely on the fact that it can't be used in ungauged regions (as state by authors in conclusions). In my opinion, this is not totally true and the use of the SCAN database is probably underutilized in this study. There is no interest to use a method which can be used only in regions where we have the truth. However, the paper can be much more improved if a last paragraph is added to discuss about the way to derive the 5 required fifth-order polynomial coefficients from climatic/soil/vegetation characteristics related to other 190 SCAN sites. Depending on the results, this publication can be useful to explain that this solution is probably (or not) the right direction to search.

The effect of soil freezing is also not addressed in the paper while there is a clear signal in humid continental pixels (Fig.6). Sudden decreases of the surface soil moisture (5cm) in winter periods (around 1/1 and 3/1 in Centralia Lake and Molly Caren for instance) are probably occurring only at the surface and not in deeper layers. This water doesn't disappear contrary to sudden decrease during summer periods. Consequently, the CDF model generate too strong decrease in winter. This point has to be discussed in the paper.

It is necessary to use a coherent formulation for the different soil moisture profile: in-situ soil moisture profile (sometime called "Profile", or "ThetaP"), the CDF profile ("ThetaP^" or "Profile (CDF)"), as well as the exponential filter ("wp^" or "profile(SWI)", I would suggest Profile(EF)).

Specific comments
1) US-SCAN network consists of over 200 soil moisture station across US but only 12 of them were used without any explanation of that choice (p.3 line 29). Authors should clarify this point.

2) "The outliers were then excluded from the analyses" (p.4 line 5). Does this mean that the station is excluded from the analysis or only the period concerned with outliers?

3) P.6 line 26. The term "reference" is somewhat confusing. Actually, the exponential filter is not used as a reference method to judge the performance of observation operators but as an alternative method. The reference dataset is in-situ soil moisture. Same remark p.9 line 2 and p. 12, line 2.

4) P.7 lines 15-20. I do agree that exponential filter (EF) provides index values (between 0 and 1) and need to be rescaled using w2max and w2min values. But this solution only required 2 parameters whereas the CDF matching procedure required a large amount of 5 cm soil moisture measurements to derive the 5 coefficient of the fifth-order polynomial procedure. The sentence seems to indicate that these 2 coefficients are a strong limitation of the EF method. Authors should reformulate this paragraph.

5) I don't think Eq. 12 is required as Eq. 11 is supposed to provide a 0-100cm soil moisture content. I expect results of each equation to be quite similar (L2(=100)»L1(=5)).

6) Section 3.2.1. It is not clear how the three correlation coefficients (0.92, 0.75 and 0.59) can be observed in Fig. 4. Please clarify

7) P.10 lines 18-19 and Figure 9. I can understand why the bias related to the exponential filter method is not equal to 0 since the 0-1 time-serie is scaled using the min and max value. However, looking at Fig. 8 (top-middle graph for instance), the green curve does not seem to be scaled to the black one. Therefore, I expect the bias (Fig. 9) to be slightly overestimated.

8) P.11 line 10. I do not agree with the sentence "that only surface (5 cm) soil moisture is needed as the input". Actually, the method also requires deeper layer soil moisture

measurements (up to 100 cm) to derive fifth-order polynomial coefficient of the CDF procedure. Please reconsider this sentence.

Technical corrections

1) Table 2: Authors should indicate in the legend which are the time-serie compared in this Table. Idem in Fig.3

2) Fig. 4, indicate that the 3 top graphs correspond to "humid continental" pixels, and idem for middle and bottom graphs.

3) Idem Fig.4 remark

4) Fig.9: Authors should better explain the meaning of "a", "b" and "ab"

5) P.5 line 17: a word is missing after "then". Probably "used" ?

6) P10. Line 30. A work is missing after "clearly". Maybe "shows"

---

## Author Comment (AC2) · 28 Jul 2017

**Responses to the comments of Dr. Yijian Zeng**

The authors developed a CDF matching-based approach to estimate the profile soil moisture. The preliminary application of the approach over different climate zones (e.g. with a certain number of in situ stations) shows the promising results. I have only two main concern as follows:

>> Thanks a lot for your comments and helpful suggestions. The followings are our point-to-point responses.

Major concern: 1. The author indicated the validation and calibration periods specifically for the current scaling approach. On the other hand, it is not clear why the certain year was selected for validation, why the certain year was selected for calibration. Please, the author clarifies, for example, why 2012, 2013 selected for validation, instead of 2014, 2015? I'm also curious how the selection of different cal/val periods will affect the results.

>> When analyzing the effects of data length on the performance of observation operators, two years was used initially as the maximum data length. The data in the 2010 and 2011 were used for calibration and that in 2012 and 2013 for validation. But we found that two years were not long enough after checking the changes of prediction accuracy with data length. Therefore, we extended the maximum data length to four years and the data in 2014 and 2015 were used. In order to minimize the workload, the data in the 2012 and 2013 remained for validation. And in the following analysis for different climates, soil moisture in 2013 (some stations used 2016 or 2010 because considerable data missing in 2013) was also used for validation.

To our knowledge, the validation of one method requires independent data which however is not necessarily later in time than calibration data. But it is interesting to test the effects of different calibration and validation data on the performance of observation operators. Therefore, we used the data in 2014 and 2015 as validation data and the other data for calibration and made recalculations. The statistics were shown below and the results showed that similar trends of statistical metrics with data length were observed as compared with Figure 3. And the two years of data length remained the optimal data duration. Furthermore, the station of Little Red Fox in the Figure 3 should be McCracken Mesa and has been corrected.

[Figure]

2. For figure 5, you can find negative NSC. There is no further discussion in the paper to explain why. Does it imply a limitation for applying the current developed CDF matching-based approach?

>> We are sorry that the caption of figure 4 is wrong and this should bring confusing. This graph shows the changes of NSC with the T parameter used in the exponential filter method in order to identify the optimal T value. The proper title should be "The changes of Nash-Sutcliffe coefficient with T parameters for nine SCAN stations in different climate regions".

For the Holden station, negative NSC was observed for all T values (from 1 to 50). According to Nash and Sutcliffe (1970) and Albergel et al. (2008), a value of 1 corresponds to a perfect match between predicted and observed data. A zero value indicates that the predictions are as accurate as the mean of the observed data, whereas a value of less than zero occurs if the observed mean is a better predictor than the model output. Therefore, negative NSC values here mean that for the Holden station, the predictions of the exponential filter mismatched the observed data.

Albergel, C., R¨udiger, C., Pellarin, T., Calvet, J.C., Fritz, N., Froissard, F., Suquia, D., Petitpa, A., Piguet, B., Martin, E.: From near-surface to root-zone soil moisture using an exponential filter: an assessment of the method based on in-situ observations and model simulations. Hydrol. Earth Syst. Sci., 12, 1323–1337, 2008.

Nash, J. and Sutcliffe, J.: River flow forecasting through conceptual models, part Ii – a discussion and principles, J. Hydrol., 10, 282–290, 1970.

Minor comments: See supplement.

1. It is recommended to also cite following recent paper on this perspective:

Yu, L., Y. Zeng, Z. Su, H. Cai and Z. Zheng (2016). "The effect of different evapotranspiration methods on portraying soil water dynamics and ET partitioning in a semi-arid environment in Northwest China." Hydrol. Earth Syst. Sci. 20(3): 975-990.

>> We agree. It has been edited in the text.

2. It is recommended to also cite following recent paper on CDF application for spatial upscaling:

Zeng, Y., Z. Su, R. van der Velde, L. Wang, K. Xu, X. Wang and J. Wen (2016). "Blending Satellite Observed, Model Simulated, and in Situ Measured Soil Moisture over Tibetan Plateau." Remote Sensing 8(3): 268.

>> We agree. It has been edited in the text.

3. Please indicated the specific period used for each SCAN station.

>> We agree. They are shown as follows.

(1) Blue Creek, Green River, McCracken Mesa: From Jan. 1, 2010 to Dec. 31, 2015; (2) Centralia Lake, Holden, Molly Caren, Perdido Riv Farms, Sevilleta, Silver City: From Jan. 1, 2013 to Dec. 31, 2015; (3) Shagbark Hills, and Youmans Farm: From Jan. 1, 2014 to Dec. 31, 2016; (4) Willow Wells: From Jan. 1, 2012 to Dec. 31, 2013, and from Jan. 1 to Dec. 31 in 2010. We check that the explanations in line 6 and 7 were wrong and they have been edited in the text.

4. Why such choice? Why not 2010, 2011, 2012, 2013 to establish the observation operators, and then 2014, 2015 for validation? Similar comments for the data length DL4 & DL5.

>> Please see our response to the major concern #1 in Page 1.

5. It is not clear if the authors applied the specific observation operator for individual climate region or not. And if it is so, how the climate regions were classified (e.g. according to which climate classification map), as well as how the observation operator for each climate region was determined (e.g. using average of SCAN station, or certain representative station for that climate zone?)

>> First, the climate regions were classified according to Köppen climate classification as indicated in Table 1. Second, we selected three primary climates in the continental USA in terms of humid continental, humid subtropical and semiarid climates. In each climate, we selected three stations as three replicates to test observation operators. We did not use all stations in a given climate because (1) a lot of stations had considerable missing data probably due to sensor malfunction especially in humid continental and humid subtropical regions, and (2) in our opinion three replicates were enough to do a reasonable test.

6. It is understandable to have an independent well-applied method to estimate the profile soil moisture. On the other hand, it is a bit awkward to call it as the reference method. As in this study, the reference data are from SCAN network, not from exponential filter, as well as the CDF based approach is not developed along the line of exponential filter.

>> We agree. Line 26 in page 6 has been changed into "…, served as an independent method to judge the performance of observation operators in different climate regions".

7. L=L1+L2?

>> L and L2 should represent one parameter, i.e., the depth of the second layer. It has been corrected in the text.

8. What is Tau? What is t?

>> According to Wagner et al. (1999) and Albergel et al. (2008), Equation (8) is the analytical solution of Equation (7). Here both tau ($\tau$) and $t$ represent time. Tau is a temporary parameter and used in order to differentiate $t$ in the integration expression in the Equation (8).

9. I am really curious how the determination of calibration and validation period will affect the results here. Please discuss a bit.

>> We agree. To our knowledge, the calibration data and validation data should be independent each other. But it is interesting to test the effects of different calibration and validation data on the performance of observation operators. Therefore, we used the data in 2014 and 2015 as validation data and the other data for calibration and made recalculations. The statistics were shown below and the results showed that similar trends of statistical metrics with data length were observed as compared with Figure 3. And the two years of data length remained the optimal data duration. Furthermore, the station of Little Red Fox in the Figure 3 should be McCracken Mesa and has been corrected.

[Figure]

10. It is not clear what are those three primary climate regions. Are these nigh stations listed also in Table 1?

>> They are humid continental, humid subtropical and semiarid climates. The nine stations including Centralia Lake, Molly Caren, and Shagbark Hills in humid continental climate; Perdido Riv Farm, Silver City, and Youmans Farm in humid subtropical climate; and Holden, Sevilleta, and Willow Wells in semiarid climate. They were all listed in Table 1.

11. In fact, the exponential filter here is an independent method to estimate profile soil moisture. It seems a bit awkward to call it a reference method, as the reference datasets are from SCAN not from exponential filter.

>> We agree. Line 26 in page 6 has been changed into "…, served as an independent method to judge the performance of observation operators in different climate regions".

12. why this negative NSC values? Does it imply a limitation for the current developed CDF-based approach?

>> We are sorry that the caption of Figure 5 was wrong. Please see our response to the second major comment in Page 1.

13. This is not appropriate. The relative high soil water content is mainly caused by the humid climate with more frequent precipitation events.

>> We agree that the relative high soil water content is because of the high precipitation in the humid climate. Here the optimal T value ($T_{opt}$) is much lower in humid subtropical climate and a lower $T_{opt}$ reflects a faster response of second-layer soil moisture (w2) to surface moisture (w1) there. Here we mean that the faster response of w2 to w1 in humid subtropical climate can be because of high hydraulic conductivity (vertical water flow velocity) due to high moisture content.

14. This is not appropriate. It is mainly due to the lack of precipitation. However, the hydraulic conductivity is not necessary lower in semi-arid regions than in humid areas.

>> We agree that the low soil moisture in semiarid region is due to the lack of precipitation and that the saturated hydraulic conductivity is not necessary lower in semiarid regions than in humid regions because saturated hydraulic conductivity is primarily controlled by soil texture. But unsaturated hydraulic conductivity is greatly affected by soil moisture content. Dry soils usually have low hydraulic conductivity because water flow in soils is very slow. Therefore, here we mean that the soils in semiarid region are dry and thus the velocity of water flow between surface and subsurface soils is low, i.e., low hydraulic conductivity. This indicates that slower second-layer response to surface infiltration, which corresponds to the higher $T_{opt}$ value in semiarid climate.

15. What is the criteria to choose calibration and validation data?

>> The primary criteria of choosing data are the completeness of the soil moisture observations. In the calibration period, the soil moisture in 2015 for the Perdido Riv Farms and that in 2014 and 2015 for the Willow Wells have a lot of missing data. In the validation period, the soil moisture in 2016 and 2010 was used for the Perdio Riv Farms and Willow Wells, respectively. This is because of considerable missing data in 2013 for these two stations. Furthermore, note that only soil moisture data in 2013 was used for validation for other stations because soil moisture data in 2012 included a lot of missing values for several stations.

16. 'high' soil hydraulic properties?

>> It should be high soil hydraulic conductivity and has been corrected in the text.

17. Please indicate climate zones in the figure.

>> We agree. It has been added in the Figure 4.

---

## Author Comment (AC4) · 15 Aug 2017

**Responses to the comments of Dr. Thierry Pellarin**

The manuscript presents a statistic approach to derive Root-zone soil moisture (0-100 cm) from near-surface soil moisture measurements (5cm). This is an important and current topic since remote sensing techniques devoted to measure soil moisture are only sensitive to the top 5 cm soil moisture and most of applications (agriculture, meteorology, hydrology) are interested in deeper soil moisture estimates. The proposed methodology consists in using a CDF-matching procedure applied to the 5 cm soil moisture data to predict 0-100 soil moisture dynamic. To my knowledge, this has not been tested in previous studies. Authors address the problem of three issues: the calibration period-length, the temporal sample, and the climatic effect on the surface/deep soil moisture link. Dataset comes from the US SCAN network. Although the subject is interesting and deserves publication in HESS, the paper presents some deficiencies. For this reason, I would recommend major revisions prior to potential publication in HESS.

>> Thank you very much for your constructive comments on our paper. Here are our point-to-point responses to all of the comments.

General comments:

The main problem of the proposed methodology rely on the fact that it can't be used inungauged regions (as state by authors in conclusions). In my opinion, this is not totally true and the use of the SCAN database is probably underutilized in this study. There is no interest to use a method which can be used only in regions where we have the truth. However, the paper can be much more improved if a last paragraph is added to discuss about the way to derive the 5 required fifth-order polynomial coefficients from climatic/soil/vegetation characteristics related to other 190 SCAN sites. Depending on the results, this publication can be useful to explain that this solution is probably (or not) the right direction to search.

>> Thanks for this valuable suggestion and we agree. It is interesting to link the polynomial coefficients with the climatic/soil/vegetation properties of the SCAN stations. In this manner, we can obtain the observation operators by using these properties. Here is our way. First, we choose the SCAN stations having at least two-year continuous soil water observations, and then establish observation operators by using the fifth-order polynomial coefficient for each selected station. Second, the

basic and easily measured environmental variables are collected for all of the selected SCAN stations. These variables include, for examples, mean annual precipitation in climate, soil texture fractions (sand, silt and clay) in soil, and aboveground net primary production in vegetation. Finally, the multivariate linear regression (MLR) model is employed to associate these environmental variables (inputs) with each respective polynomial coefficient. And then the five MLR models can be used to estimate the polynomial coefficients. The text has been edited.

The effect of soil freezing is also not addressed in the paper while there is a clear signal in humid continental pixels (Fig.6). Sudden decreases of the surface soil moisture (5cm) in winter periods (around 1/1 and 3/1 in Centralia Lake and Molly Caren for instance) are probably occurring only at the surface and not in deeper layers. This water doesn't disappear contrary to sudden decrease during summer periods. Consequently, the CDF model generate too strong decrease in winter. This point has to be discussed in the paper.

>> Thanks for this valuable suggestion and we agree. Yes, we notice that surface soil moisture and the CDF-matching predictions suddenly decreased in freezing periods under the humid continental climate. This could suggest that soil freezing influences the surface-profile soil moisture relations and the prediction accuracy of the CDF-matching model. Therefore, we suggest that the soil moisture observations during soil freezing period in cold climates should be excluded because they do not represent the true soil moisture content. The text has been edited.

It is necessary to use a coherent formulation for the different soil moisture profile: in-situ soil moisture profile (sometime called "Profile", or "ThetaP"), the CDF profile ("ThetaP^" or "Profile (CDF)"), as well as the exponential filter ("wp^" or "profile(SWI)", I would suggest Profile(EF)).

>> We agree. They have been edited in the manuscript.

Specific comments
1) US-SCAN network consists of over 200 soil moisture station across US but only 12 of them were used without any explanation of that choice (p.3 line 29). Authors should clarify this point.

>> We agree. There are two primary grounds for our selection. First, it is because a lot

of stations contained considerable missing data at one or several depths especially in humid continental and humid subtropical climates. Second, three stations were used as three replicates for each climate and also for optimizing data resolution and length because three replicates generally are enough in statistics.

2) "The outliers were then excluded from the analyses" (p.4 line 5). Does this mean that the station is excluded from the analysis or only the period concerned with outliers?

>> It means that only the period concerned with outliers were excluded.

3) P.6 line 26. The term "reference" is somewhat confusing. Actually, the exponential filter is not used as a reference method to judge the performance of observation operators but as an alternative method. The reference dataset is in-situ soil moisture. Same remark p.9 line 2 and p. 12, line 2.

>> We agree. The term "reference" in these lines has been changed into "independent".

4) P.7 lines 15-20. I do agree that exponential filter (EF) provides index values (between 0 and 1) and need to be rescaled using w2max and w2min values. But this solution only required 2 parameters whereas the CDF matching procedure required a large amount of 5 cm soil moisture measurements to derive the 5 coefficient of the fifth-order polynomial procedure. The sentence seems to indicate that these 2 coefficients are a strong limitation of the EF method. Authors should reformulate this paragraph.

>> First of all, the exponential filter method also requires preceding soil moisture data in surface and subsurface layers to obtain the optimal $T$ parameter. The predicted second-layer soil moisture ($SWI_2$) by the exponential filter method was rescaled by using w2max and w2min just in order to compare with the CDF-matching method which used original moisture values. In most cases (e.g., Wagner et al., 1999; Albergel et al., 2008; Brocca et al., 2011), however, the scaled soil moisture (0-1) is usually used for the exponential filter method and is not needed to do rescaling. Therefore, the two parameters of w2max and w2min would be not a limitation for the EF method in these cases. The text has been edited for clarity.

*Albergel, C., et al.: From near-surface to root-zone soil moisture using an exponential filter: an assessment of the method based on in-situ observations and model simulations, Hydrol. Earth Syst. Sci., 12, 1323–1337, 2008.*

*Brocca, L., et al.: Soil moisture estimation through ASCAT and AMSR-E sensors: an intercomparison and validation study across Europe, Remote Sens. Environ., 115, 3390–3408, 2011.*

*Wagner, W., Lemoine, G., and Rott, H.: A method for estimating soil moisture from ERS scatterometer and soil data, Remote Sens. Environ., 70, 191–207, 1999.*

5) I don't think Eq. 12 is required as Eq. 11 is supposed to provide a 0-100cm soil moisture content. I expect results of each equation to be quite similar (L2(=100)»L1(=5)).

>> In the exponential filter method, the second layer refers to the soil depth in the 5-100 cm, but in the CDF-matching method the profile refers to the top 100 cm. Therefore, in our opinion, it is needed to use the Eq. 12 to get the profile soil moisture for the exponential filter although the results are very similar.

6) Section 3.2.1. It is not clear how the three correlation coefficients (0.92, 0.75 and 0.59) can be observed in Fig. 4. Please clarify

>> The correlation coefficient equals to the maximum cross correlation value. The three coefficients of 0.92, 0.75 and 0.59 are the arithmetical means of the maximum cross correlation coefficients in the humid subtropical, humid continental and semiarid climates, respectively. It has been clarified in the text.

7) P.10 lines 18-19 and Figure 9. I can understand why the bias related to the exponential filter method is not equal to 0 since the 0-1 time-series is scaled using the min and max value. However, looking at Fig. 8 (top-middle graph for instance), the green curve does not seem to be scaled to the black one. Therefore, I expect the bias (Fig. 9) to be slightly overestimated.

>> We are not sure whether the non-zero MBE values are necessarily related to the rescaling of the predictions ($SWI_2$) by using the max and min values for the exponential filter. The difference between the predicted (green line) and observed (black line) profile moisture content for the EF method might be resulted from the assumption that the water flux between surface and subsurface layers is proportional

to the difference in soil moisture between these two layers. In fact, this relationship should be nonlinear with soil moisture content. Furthermore, this assumption may result in relative low Nash-Sutcliffe coefficient (< 0.7 for the majority of stations and even negative) even at the optimal $T$ ($T_{opt}$) in calibration period (Figure 5). This means that there is a mismatch between the predicted and observed profile moisture content (Figure 6-8).

8) P.11 line 10. I do not agree with the sentence "that only surface (5 cm) soil moisture is needed as the input". Actually, the method also requires deeper layer soil moisture measurements (up to 100 cm) to derive fifth-order polynomial coefficient of the CDF procedure. Please reconsider this sentence.

>> Here we mean that as soon as the observation operators are built only surface soil moisture is needed as the input to predict profile soil moisture. We have rewritten this sentence for clarity.

Technical corrections

1) Table 2: Authors should indicate in the legend which are the time-serie compared in this Table. Idem in Fig.3

>> We agree. Soil moisture observations in the years of 2014 and 2015 were used for the Shagbark Hills and Sevilleta stations and that in the years of 2013 and 2014 for the Perdido Riv Farms station. It has been edited.

2) Fig. 4, indicate that the 3 top graphs correspond to "humid continental" pixels, and idem for middle and bottom graphs.

>> We agree. It has been edited.

3) Idem Fig.4 remark

>> We agree. It has been edited.

4) Fig.9: Authors should better explain the meaning of "a", "b" and "ab"

>> We agree. Here the Least Significant Difference (LSD) method was used to do Post Hoc multiple comparisons for the one-way ANOVA. First, one statistical metric (RMSE, $R^2$ or MBE) was ordered from the highest to the lowest by value among the three climates. Next, the climate having the largest value (C1) was labeled with "a".

Then C1 was compared with the climate having the medium value (C2). If they were significantly different ($P<0.05$), the C2 was labeled with "b", and then C2 was compared with the climate with the lowest value (C3). Similarly, if they were significantly different ($P<0.05$), the C3 was labeled with "c". However, if C1 and C2 were not statistically ($P<0.05$) different, C2 was labeled with "a" as well and C1 would further compare with C3. If C3 and C1 were significantly ($P<0.05$) different, C3 was labeled with "b". In this case, C3 should be also compared with C2. If they were significantly ($P<0.05$) different, the comparison ended; or else, C2 was further labeled with "b" and then the label of C2 became "ab".

In fact, we had given a short explanation in the title of Figure 9. Because the above explanations can be accessible in statistical textbooks, they would not be indicated in the revisions.

5) P.5 line 17: a word is missing after "then". Probably "used" ?
>> We agree. It has been edited.

6) P10. Line 30. A word is missing after "clearly". Maybe "shows"
>> We agree. It has been edited.